

# Enhancing Flood Hazard Estimation Methods on Alluvial Fans Using an Integrated Hydraulic and Geological and Geomorphological Approach

Zeinab Mollaei[1], Kamran Davary[2], Seyed Majid Hasheminia*[2], Alireza Faridhosseini[2] and Yavar Pourmohamad[3]

[1] Graduated MSc Student of Water Science & Engineering Dept., College of Agriculture, Ferdowsi University of Mashhad, P. O. Box: 91775-1163, Mashhad, Iran.
[2] Faculty member of Water Science & Engineering Dept., College of Agriculture, Ferdowsi University of Mashhad, P. O. Box: 91775-1163, Mashhad, Iran.
[3] PhD student of Water Science & Engineering Dept., College of Agriculture, Ferdowsi University of Mashhad, P. O. Box: 91775-1163, Mashhad, Iran

*Correspondence to*: Seyed Majid Hasheminia (hasheminia@ferdowsi.um.ac.ir)

**Abstract.** Due to the uncertainty concerning the location of flow paths on active alluvial fans, alluvial fan floods could be more dangerous than riverine floods. FEMA used a simple stochastic model named FAN in this regard for many years. In the last decade, this model has been criticized as a consequence of developing complex computer models. Instead, more recent hydraulic models with capability of avulsion simulation were recommended. This study was conducted on three alluvial fans located in the northeast and southeast part of Iran using combination of FAN model, FLO-2D model and geomorphological information. Initial stages included three steps: a) identifying the alluvial fans' landforms, b) determining the active and inactive areas of alluvial fans, and c) delineating 100-year flood within these selected areas. This information was used as an input in the mentioned three approaches of: I) FLO-2D model, II) geomorphological method, and III) FAN model. Thereafter, the results of each model were obtained and Geographical Information System (GIS) layers were created and overlaid. Afterwards, using a scoring system, the results were summarized and then the integrated method was compared with the three approaches.

## 1 Introduction

Alluvial fan floods are considered to be serious hazards, since the flooding that emerges from the upstream of an alluvial fan, moves fiercely toward the downstream while carrying a large portion of substrate load and debris (King and Mifflin, 1991). Moreover, alluvial fans are made of larger size sediments without cohesive material that lies on a steep slope. These two characteristics could lead to "avulsion", formation of new channels during flooding events, which causes major flow path displacement (Blair and McPherson, 2009). Estimation of flood hazards on alluvial fans has been a major dispute among hydrologists for many years (Bedrossian et al., 2014). In fact, due to "avulsion" the characteristics of alluvial fan flooding are more important than other flooding events, which could consequently increase the risk of damage (Lancaster et al., 2012). As stated by National Research Council: "the area of deposition on an alluvial fan shifts with time, but the next episode of flooding is more likely to occur where the most recent deposits have been laid down, where deposits of greatest antiquity occur" (National Research Council, 1996).



Utilizing the alluvial fan characteristics, United States Federal Emergency Management Agency (FEMA) developed a method to assess the flood risk of alluvial fans. The FEMA guidelines allow a number of delineation methodologies that include geomorphic method, one and two-dimensional fixed bed hydraulic modeling, and composite methods that combine engineering and geologic approaches (Federal Emergency Management Agency, 2003a).The first attempt to

address the alluvial fan flood complexities was performed by Dawdy (1979), who developed a probability-based model. The model was based on a mathematical formulation that was developed after a series of catastrophic alluvial fan floods and debris flows in the 1970s. FEMA correctly recognized that riverine floodplain delineation techniques did not adequately depict the flood hazards on active alluvial fans, and adopted Dawdy's equations to better describe the flood risks associated with non-riverine processes such as avulsions, high rates of sediment transport, and net

aggregation (Dawdy, 1979). FEMA applied this approach directly in a number of alluvial fan floodplain delineation studies in the 1980s, and thereafter, the FAN model was developed (Federal Emergency Management Agency and Administration, 1990). FAN model is a DOS based software package that uses Dawdy's basic equations, as well as a modification proposed by DMA Consulting Engineers (1985), to predict flow depths and velocities on alluvial fans on a regular basis (French et al., 1993).

During the last decade, FAN model has been profoundly criticized by some researchers (Field and Pearthree, 1992; French, 1992; French et al., 1993; Fuller, 2013, 1990; House et al., 1991; Pearthree et al., 1992) since it does not consider the physical characteristics of alluvial fans. Despite of all criticizes raised about this model; it still can be a useful tool to delineate alluvial fan flooding. In addition, it is a simple model which could predict avulsion. According to this model, the areas subjected to alluvial fan flooding are assumed to have an equal width (Federal Emergency

Management Agency and Administration, 1990). This width is referred to as the contour width, since it is measured along a contour. However, hydraulic models fill the sinks based on topographic maps, and depth and velocity are just computed as a point. However, in order to interpret the flood insurance areas, hazard zones are required and not just hazard points (Gallien, 2016).

Alluvial fan flows are two-dimensional, therefore, the application of one dimensional flow hydraulic models on

alluvial fans could have the following limitations : I) difficulties and inaccuracies in determining flow direction on floodplains; II) the lack of flood volumes in determining flooding boundaries, and III) the problem with the uncertainty nature of flood discharge calculation (Volker et al., 2007; Yunsheng, 2009). Marchi et al. (2010) used an integrated approach to analyze the hydro-geomorphic processes and their interactions with torrent control works and applied it to a large alluvial fan in the southern Carnic Alps (northeastern Italy). Their study encompassed field observations,

interpretation of aerial photographs, analysis of historical documents, and numerical modelling of debris flows (Marchi et al., 2010).

In addition, it has been found that geological and geomorphological data have great impacts on estimation of avulsion intensities (Fuller, 2012). Studies conducted on 100-year floods in alluvial fans using the mentioned methods, emphasize the fact that formulating a framework for the geological-geomorphological features of the area can

profoundly improve our understanding about their role in flood risk assessment (House, 2005). In another study Pelletier et al. used integrated geologic maps, geomorphic analysis and a two-dimensional hydrodynamic model (LISFLOOD) to assess flood hazards (Pelletier et al. 2005). In a study, a multi-criteria index to assess flood hazard





areas in a regional scale was introduced. Accordingly, a flood hazard index (FHI) was proposed and a spatial analysis in geographic information system (GIS) environment was applied for the estimation of its value (Kazakis et al., 2015). In another study, framework was presented for mapping potential flooding areas integrating GIS, fuzzy logic and clustering techniques, and multi-criteria evaluation methods by Papaioannou et al. (2015). To sum up, the geological

maps could be a powerful tool in better analyzing avulsion due to incorporating the effects of erosion and sedimentation. In fact, using geomorphological analysis can provide: I) calibration or verification of hydraulic modeling results, II) a context for understanding the basic system processes, and III) an understanding of the type of processes that has been occurred from the past up to present.

The aim of this study was to examine the applicability of overlaying three layers of land/ground susceptible to erosion,

water erosion potential, and hydraulic flooding zones in alluvial fans. Ferizy and Ardak alluvial fans in Khorassan Province and Sarbaz fan in Sistan & Baloochestan Province, which all are located in arid regions of eastern part of Iran were considered for this study. In this paper, anew method was introduced and applied for each data layer and the results were discussed.

## 2 Methods and Materials

In this study, a simple flowchart was followed to achieve the final flood hazard map, as illustrated in Fig. 1. The proposed model needs four input data in order to obtain a flood hazard map, which are Q100hydrograph, topographic map, normalized difference vegetation index (NDVI) and field investigations.

Five steps have been followed to achieve this goal, which areas follows:

1. FAN model was established using the 100 year return period hydrograph and the average slope which was
extracted from topographical maps.

2. FLO-2D model was executed by using the 100 year return period hydrograph and topographical maps.

3. Active and inactive areas were distinguished by considering topographical maps, NDVI and field investigations.

4. Thereafter, all the models, results and field investigations were georeferenced and overlapped in GIS as separated layers and each layer was appointed a score using a scoring system from zero to 1.5.

5. Finally, the score of layers was multiplied and afterwards the pixels with the highest value and pixels with zero score were considered to be the highest and the lowest hazard zones, respectively.

### 2.1 Input Data

### 2.1.1 Hydrograph (Q$_{100}$)

Thirty-five years of hydrometric data was obtained from Water Authority Company for each area. Afterwards the 100-
year return period hydrograph were developed (Q100) based on collected data for all three fans in a spreadsheet. These hydrographs were used in FAN model and FLO-2D as input data. Figure 2 (a, b, and c) illustrates the Q100 hydrographs for Ardak, Ferizy and Sarbaz fans, respectively.



### 2.1.2 Topographic Map

Georeferenced elevation maps were downloaded in Tiff files format from USGS website for all three fans. Afterwards, using GIS tool, the average slope map was derived from elevation map and used as an input in FAN model. Georeferenced elevation maps were used as FLO-2D model input; however, contour maps were the primary input of

this model. The process of converting elevation maps to contour maps was performed in FLO-2D model, therefore, no extra calculations were needed. Figure 3 (a, b, and c) shows the elevation map for Ardak, Ferizy and Sarbaz fans, respectively.

### 2.1.3 NDVI

To cover all three study areas, two Landsat 8 OLI Images (path/row: 159/35 and 156/42) on May 13 for Ardak and

Ferizy fans and May 24 for Sarbaz fan were downloaded from the United States Geological Survey (USGS) website. These two images were clipped to alluvial Fan borders.

To calculate NDVI, first reflectance of each of band 4 and 5 (red and infrared) were calculated based on Landsat 8 user handbook (Zanter, 2016) and then the following equation was applied on bands 4 and 5 reflectance (Li et al., 2013).

$NDVI= ((\rho\_NIR-\rho\_Red ))/((\rho\_NIR+\rho\_Red ) )$        (1)

Where, $\rho\_NIR$ and $\rho\_Red$ are reflectance of near infrared band and reflectance of red band, respectively.

### 2.1.4 Field Investigations

On a trip to Ardak and Ferizy fan in May 20 and on another trip to Sarbaz fan on May 25of 2015, some geomorphological features for delineation of active and inactive areas were assessed. These information include:

slope, drainage patterns, topographic contour, superficial characteristics, desert pavement, desert varnish, color and distinctive vegetation (Elkhrachy, 2015). The reason for this investigation was to find out active and inactive areas and validate the satellite images. The observation conclusions are presented in Table 1.

Landsat 8 aerial photographs was used to determine active or inactive areas, which were downloaded from USGS website. The evaluation was done by using the surface color and vegetation detection in photos. A dark surface color

indicated an inactive zone, while lighter surfaces represented more active areas (Lohani et al., 2006).

### 2.2 Models

### 2.2.1 FEMA's FAN Model

As stated before, the FEMA method uses Dawdy's theory to delineate alluvial fan flooding to develop the FAN model

(Federal Emergency Management Agency and Administration, 1990). Based on Dawdy's theory the flood channel occurs randomly in active alluvial fans. Therefore, each point in an active area of a fan has similar chance to become



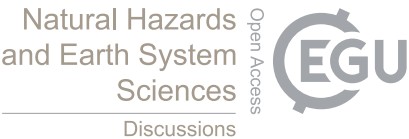

a new channel as other points; so each point has the tendency to undergo flood. Hence, the probability of flood incidence in this model can be determined as follows (for more information see FAN model manual):

$$P(H = 1) = \int_{q_0}^{\infty} P_{H/Q}(1/q) f_Q(q) dq \quad (1)$$

(2)

$$H = \begin{cases} 1 & \text{if the location is inundated} \\ 0 & \text{if the location is not inundated} \end{cases}$$

Where, Q is a random variable denoting the magnitude of the flood, $p\_(H\!/\!(Q\ (1/q)))$ is the conditional probability that a location will be inundated, given that a flood of magnitude q is occurring and fq(Q)is probability density function (PDF) defining the likelihood that a flood of a magnitude between q and q+dq will occur in any given year.

Based on the assumptions of this model all areas of the alluvial fan could be subjected to flooding and there is a fixed relation between flooding depth and discharge (Federal Emergency Management Agency, 2003b; Federal Emergency

Management Agency and Administration, 1990). The flood hazard areas on an alluvial fan are identified as Zone AO. It is defined as the flood insurance rate zone that corresponds to the areas of l00-yr shallow flooding (usually sheet flow on sloping terrains), where average depths are between 0.3048 m and 0.9144 m. The flood hazard area on an alluvial fan is subdivided into separate AO zones with similar flooding depths (water depth plus velocity head) and velocities (Zhao and Mays, 1996). Figure 5 is an example of AO zones according to depth and velocity (Federal

Emergency Management Agency and Administration, 1990).

### 2.2.2 FLO-2D Model

FLO-2D is a two dimensional flood routing hydraulic model which is commonly used by floodplain engineers to delineate flood hazard maps, implement floodplain zoning, and design flood mitigation schemes (Mollaei et al., 2016; Yunsheng, 2009). In this model full dynamic wave equation and central finite-difference routing scheme with eight

potential flow directions are used to predict the progression of a flood hydrograph over a system of square grid elements (O'Brien and Gonzalez-Ramirez, 2011). This model is characterized by some substantial advantages (Fuller, 2010):

I)  Runoff can flow anywhere along the dominant boundary, not only the concentration point,

II)  Peak discharge can be generated anywhere within the model domain, not just at the concentration route,

III)  In adjacent alluvial fans, the flow can be easily modelled along unconfined boundaries,

IV)  Watershed parameters were distributed over each small grid cell rather than being lumped over large sub-basins, and

V)  There is no need to estimate hydrologic routing parameters or average the hydraulic routing cross sections, since routed hydrographs are inherited in the model.

The FLO-2D-based model revealed that floods are not transferred via a single channel at fan evaluation sites and flow path locations could be predicted if floods have minimal sediment transport and relatively constant topography (Fuller 2010).

This model is sensitive to grid size and topographical data. To increase the accuracy of the model, small size grids and topographical data with minimum distance between contour lines should be used (Fuller, 2012). In this study, FLO-





2D model was used only as a hydraulic model, so a precision of 20-meter grid size and 3-meter topographical data were chosen, while the values of Manning coefficients were determined according to the land use information.

### 2.2.3 Geomorphological Analysis

Geomorphological features for delineation of active and inactive areas include: slope, drainage patterns, topographic
map, superficial characteristics, desert pavement and varnish, color and distinctive vegetation (Field and Pearthree, 1992; Fuller, 2013; House, 2005; Lancaster et al., 2012).

For this purpose aerial photographs were used to distinguish active or inactive areas (Federal Emergency Management Agency, 2003a). This was performed by using the surface color and vegetation detected in photos. A dark surface color is an indication of an inactive zone, while lighter surfaces represent areas that are more active. The density of
natural vegetation provides useful insights about inactiveness of an area. For instance, areas with annual plants could be considered as an active areas, while, areas with perennial plants were considered as inactive. In addition, observation of human activities or weathering footprints (desert pavement, desert varnish and limestone cracks or grooves) can provide useful information about the extension of inactiveness of an area. Finally, distributary, braided and branching drainage patterns are characteristics of active areas, whereas tributary patterns are an indicator of
inactive areas. For this purpose, the 2015 Arial images were downloaded from USGS website.

### 2.3 Overlaying the GIS Layers

Hazardous zones in FLO-2D and FAN models and geomorphological characteristics studies were defined based on their depths, velocities and areas of activeness. These criteria are presented in Table2.Note that FLO-2D model classified inundated areas as hazardous zones based on criteria presented in Table 2. This model assumes that the rest
of the areas as being out of flooding risk.

### 2.4 Study Area

This study was performed on three separated alluvial fans (Ferizy, Ardak and Sarbaz) in Iran, which are located in eastern and southeastern part of Iran. Ferizy fan is located on the northern piedmont slopes of the Binalood Mountains in Khorasan province. Currently, Ferizy fan is under development by agricultural, residential and also industrial
projects. Chenaran industrial area is situated right below the hydrographic apex (Mollaei et al., 2016).

Ardak fan, located on the southern piedmont slopes of the Hezarmasjed Mountains, which is also located within the Khorasan province. This alluvial fan is a result of Ardak river aggradations. Ardak fan has been subjected to extensive agricultural and residential development in the area, due to its fertile soil (Mollaei et al., 2016).

Sarbaz fan is located on the southern piedmont of the Sistan Mountains in Sistan & Baloochestan province, which is
located in the southeastern part of Iran. Due to its arid climate, this fan suffers from low average precipitations, high temperatures, low humidity, poor vegetation index, high aggradations and high volume of runoff and wind erosion. Sarbaz fan has been formed as a consequence of Sarbaz River aggradations, and unlike Ardak and Ferizy fans, it has not gone under development. Historic data indicate occurrence of heavy floods and serious destructions in this area, which has been a consequence of flash floods.



### 3 Results and Conclusions

#### 3.1 FAN Model Results

FAN model output was a text file, which was interpreted in ArcGIS as a shape file for each fan. FAN model results are illustrated in Fig. 6 (a, b, and c) for Ferizy, Ardak and Sarbaz fans, respectively. The results are shown in different colors, which correspond to different depths (m) and velocities (m.s-1) as explained in Table 3.

#### 3.2 FLO-2D Model Results

FLO-2D model outputs were a shape file with spatial resolution of 30m. This model produced three different results of depth (m), velocity (m.s-1) and hazard map (low, medium and high). Its criteria is also explained in Table 3. Figure 7 (a, b, and c) illustrates the depth results of FLO-2D model for Ferizy, Ardak and Sarbaz fans, respectively. Also Fig. 8 (a, b, and c) illustrates the velocity results of FLO-2D model for the mentioned fans.

#### 3.3 Geomorphological Analysis Results

A comparison of the geomorphological characteristics of the three fans showed that while Ardak fan had the greatest area, slope and discharge, the Ferizy fan had the lowest discharge (Table 3). In Table 4, a comparison of three fans with regard to their effective characteristics on flood risk hazards is presented.

As it can be seen from Table 4, Sarbaz fan is characterized by its high velocity, low channel stability and low development. Ardak fan has steep slopes, high peak discharge and large drainage area. Ferizy fan does not have any specific characteristics as compared to Sarbaz and Ardak fans, except for the low channel stability and lower flood risks. The geomorphic map of Ardak fan shows that with the exception of main channel, other parts of the alluvial fan are developed by agricultural and residential activities. Urbanization of any part of the alluvial fan would create a resistance to abrupt changes of flow direction from its main channel, while the agricultural lands, especially those under the annual crops, would intensify this phenomenon even on the inactive areas of the fan. Obviously, orchards or woodlands are less susceptible to this event as compared to farmlands.

#### 3.4 Integrated Results

By integrating, the results of these three approaches and using the scoring system, each fan was classified into four categories of flood hazard: very high, high, medium, low and very low. Figure 9 (a, b and c) shows the final flood hazard map for Ferizy, Ardak and Sarbaz fans, respectively.

As shown in Fig. 9 (a, b and c) the flood hazard area, apart from their distance from apex of the fan, is also defined based on their distance from riverbanks. In other words, the areas in, which the river channels are located or previously used to be channels, are categorized as high hazardous zones.

#### 3.5 Conclusions

Table 5 summarizes all the pros and cons about of FLO-2D and FAN models, the geomorphological approach, and the suggested integrated model.





It can be concluded that the integrated method has the superiority of projecting alluvial fan flood hazards, which is the primary goal of such studies.

For further studies, it is suggested that HEC-RAS model be used along with FAN model and geomorphological approach and the pertaining results be validated by executing FLO-2D model utilizing full input data.

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



**Table 1. Observation conclusions from field investigations.**

|  | *Fan* | *Location* | *X* | *Y* |
|---|---|---|---|---|
| *1* | Ferizy | Apex | 36.54889 | 59.09964 |
| *2* | Ferizy | Jam Ab | 36.57275 | 59.10208 |
| *3* | Ferizy | Industrial Area of Chenaran | 36.57275 | 59.10208 |
| *4* | Ferizy | Farms | 36.60528 | 59.10227 |
| *5* | Ferizy | Ring road | 36.61469 | 59.11415 |
| *6* | Ardak | Apex | 36.74175 | 59.39865 |
| *7* | Ardak | Farms | 36.73803 | 59.39892 |
| *8* | Ardak | Rural area | 36.73184 | 59.39389 |
| *9* | Sarbaz | Apex | 26.20120 | 61.73763 |
| *10* | Sarbaz | Inactive area | 26.18299 | 61.74807 |
| *11* | Sarbaz | Suldan village | 26.15361 | 61.78557 |

**Table 2. Flood criteria to delineate hazard map.**

| Hazardous | Velocity (m.s⁻¹) | | Depth (m) | | Geomorphological Characteristics |
|---|---|---|---|---|---|
|  | **FAN Model** | **FLO-2D Model** | **FAN Model** | **FLO-2D Model** |  |
| **Very High** | V>2.0 | V>1.5 | D>1.0 | D>1.0 | Active |
| **High** | 1.5-2 | 1.2-1.5 | 0.5-1.0 | 1.2-1.5 | Active |
| **Medium** | 1.3-1.5 | 1.0-1.2 | 0.3-0.5 | 1.0-1.2 | Active/Inactive |
| **Low** | 1.0-1.3 | 0.5-1.0 | 0.15-0.3 | 0.5-1.0 | Inactive |
| **Very Low** | V<1.0 | 0.0-0.5 | D<0.15 | 0.0-0.5 | Inactive |

**Table 3. The characteristics of three alluvial fans.**

| Characteristic | Ferizy | Ardak | Sarbaz |
|---|---|---|---|
| Watershed area (apex)(km²) | 283 | 497 | 72 |
| Alluvial fan area (km²) | 155 | 436 | 70 |
| Alluvial fan slope (%) | 1.04 | 1.83 | 1.03 |
| Elevation (m) | 1192.7 | 1284.8 | 267.5 |
| Temperature (°C) | 17 | 17 | 32 |
| Precipitation (mm) | 167 | 166 | 96 |
| $Q_{100}$ at apex (m³/s) | 80.8 | 221.4 | 108 |
| Fan profile shape | Concave | Concave | Concave |
| Drainage Pattern | Channelized Tributary | Channelized Tributary | Channelized Distributary |





**Table 4. Characteristics of three fans as regard to their flood risk hazards (velocity, V, and depth, D, are in m.s$^{-1}$ and m, respectively).**

|  | Ardak | Ferizy | Sarbaz |
|---|---|---|---|
| NDVI | 0.115 to 0.71 | -0.157 to 0.768 | -0.161 to 0.427 |
| FLO-2D | 0 <V< 1.5<br>0 <D< 1.5 | 0<V< 1.5<br>0<D< 1.5 | 0<V< 1.5<br>0<D< 1.5 |
| FAN | 0 <V< 2<br>0.15<D< 1 | 0<V< 2<br>0.15<D< 1 | 0<V< 2<br>0.15<D<1 |
| Active / Inactive | 0 to 1 | 0 to 1 | 0 to 1 |
| Final Score | 0 to 3.195 | 0 to 3.456 | 0 to 1.921 |

**Table 5. Pros and Cons of three models and the suggested integrated model**

| Characteristics \ Models | FAN | FLO-2D | Geomorphological Approach | Suggested Integrated Model |
|---|---|---|---|---|
| Depth | ✓ | ✓ | - | ✓ |
| Velocity | ✓ | ✓ | - | ✓ |
| Active/Inactive | - | - | ✓ | ✓ |
| Fine Resolution Results | - | ✓ | - | ✓ |
| Required Input Data | Low | High | Low | Low |
| Degree Simplicity | Simple | Complex | Complex | Simple |



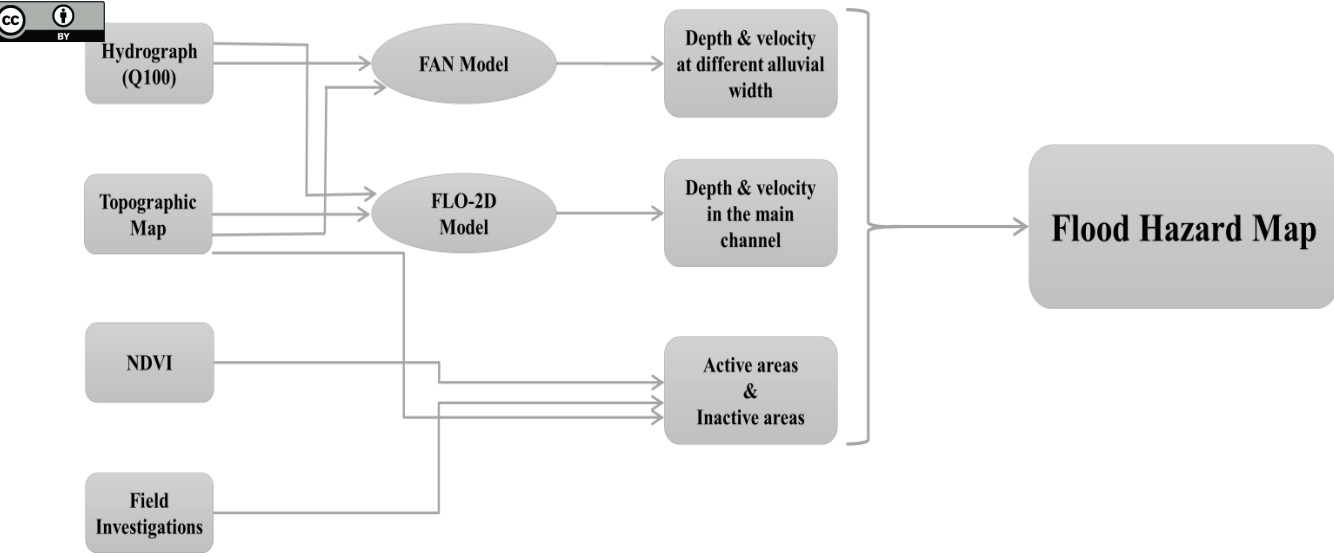

**Figure 1. The proposed flow chart to achieve flood hazard map.**





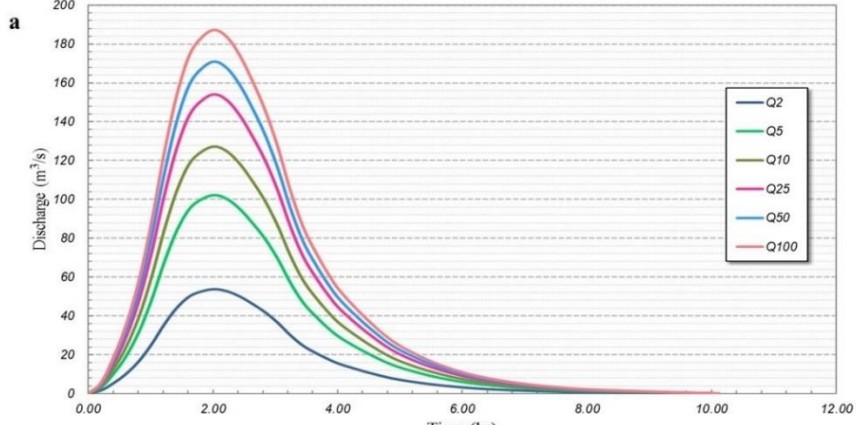

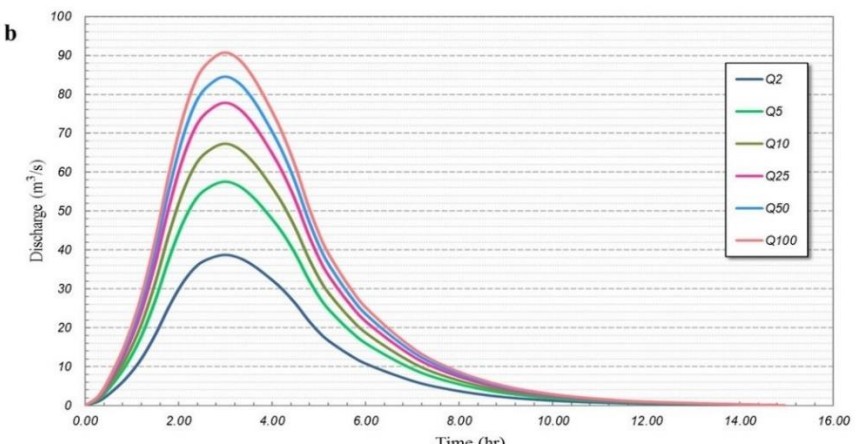

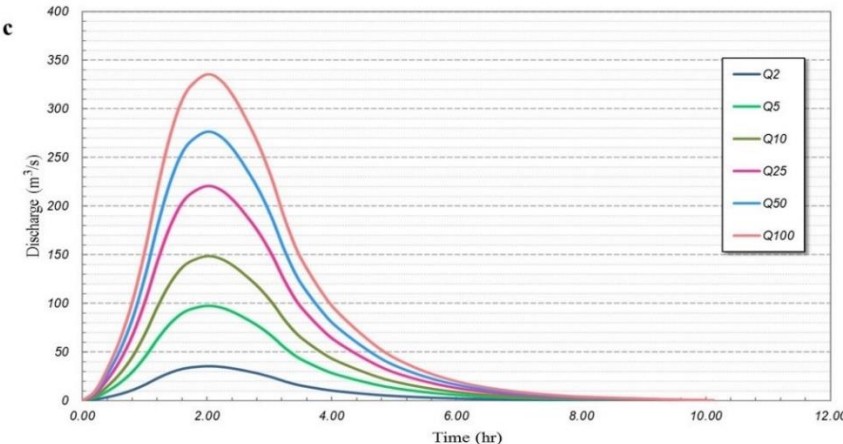

**Figure 2.** $Q_{100}$ for a) **Ardak fan,** b) **Ferizy fan and** c) **Sarbaz fan.**





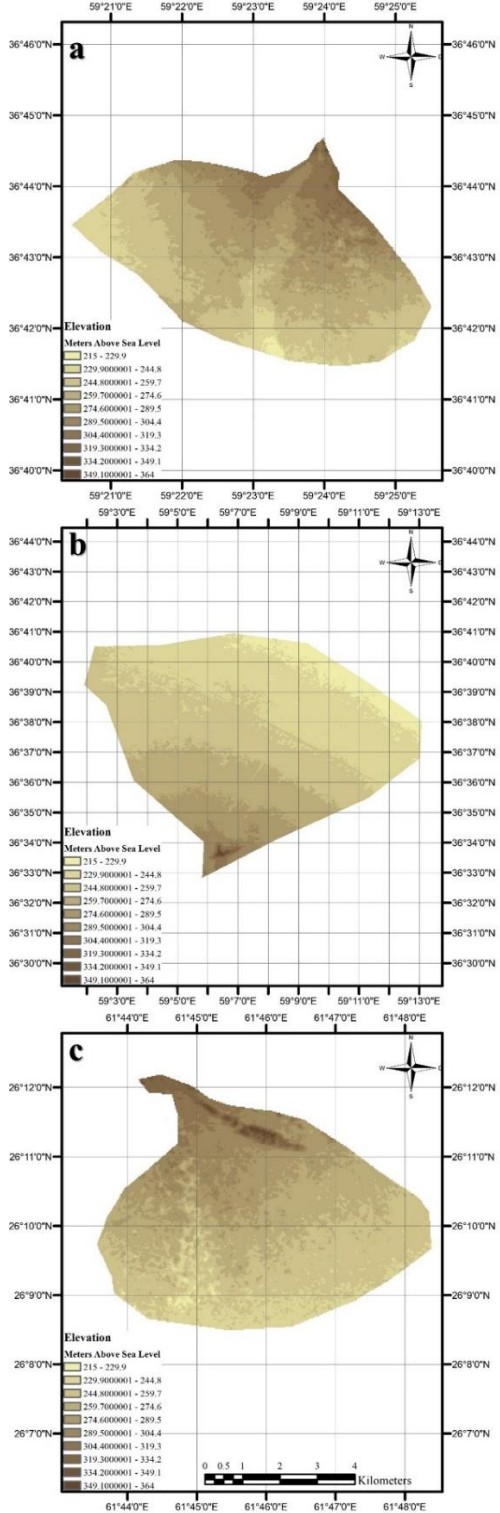

**Figure 3. Land elevation from Aster dataset for a) Ardak fan, b) Ferizy fan and c) Sarbaz fan.**

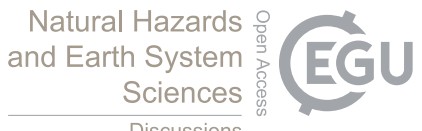
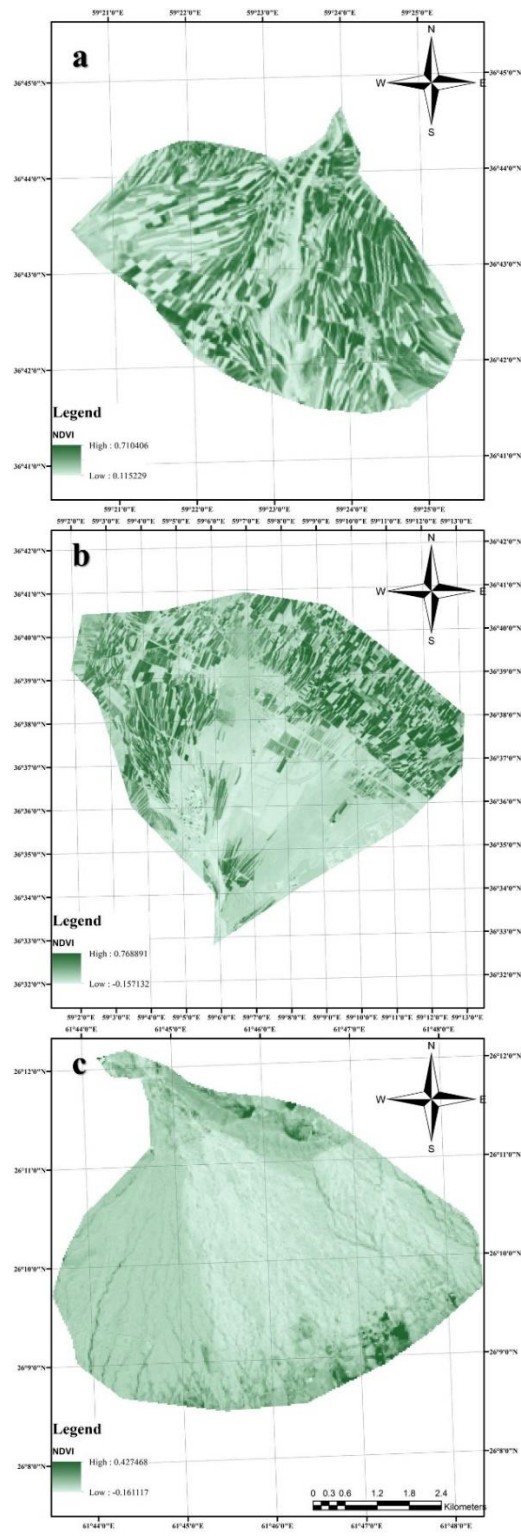

**Figure 4. Calculated NDVI for a) Ardak fan, b) Ferizy fan and c) Sarbaz fan.**



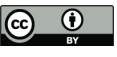

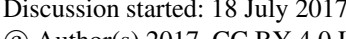

**Figure 5. Examples of AO zones according to depth and velocity in FAN model (Federal Emergency Management Agency and Administration, 1990).**





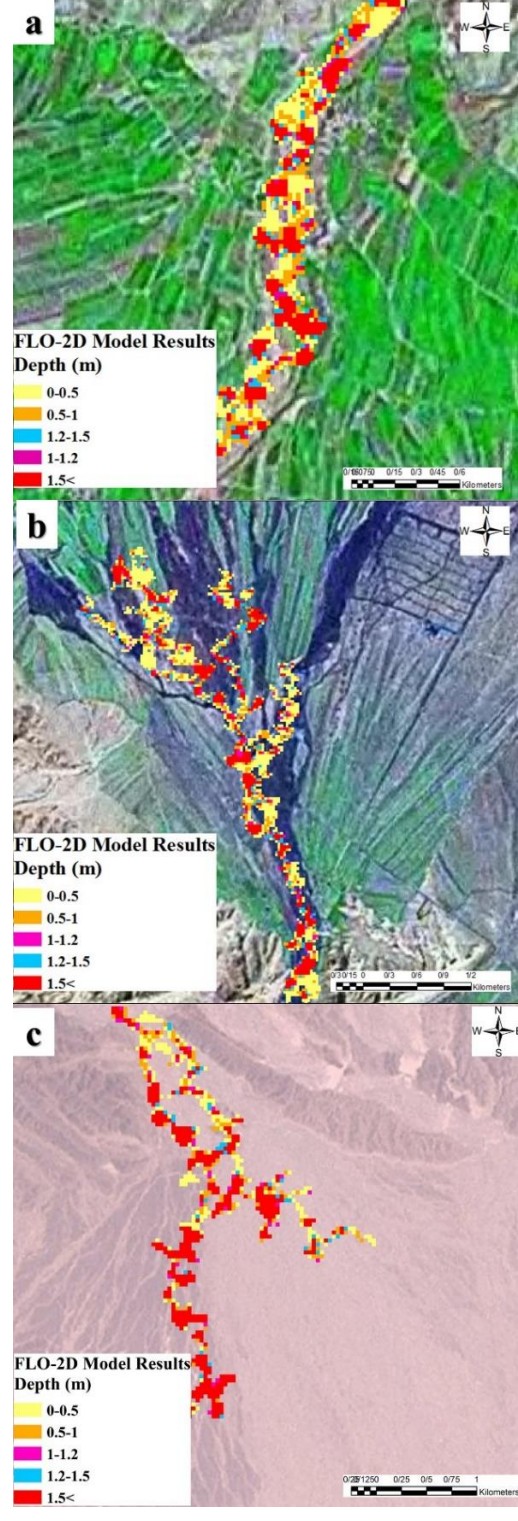

**Figure 6. FAN model results for flood depths and velocities in A) Ferizy fan, B) Ardak fan and, C) Sarbaz fan.**




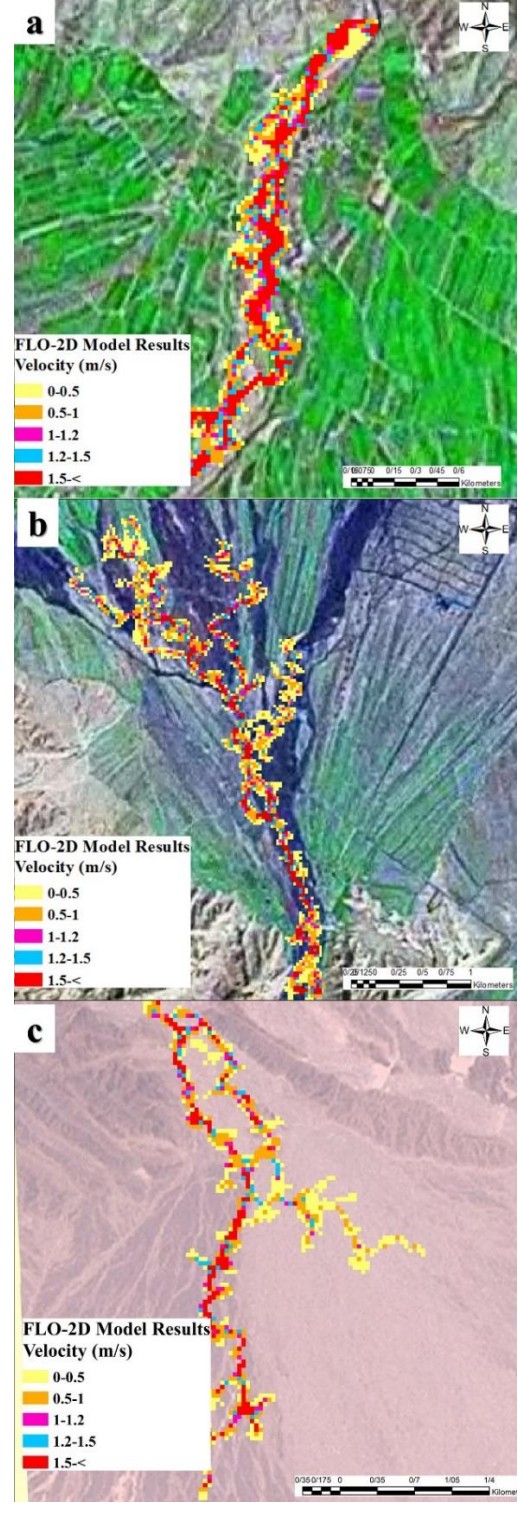

**Figure 7. Flood depth results of FLO-2D for A) Ardak fan, B) Ferizy fan, and C) Sarbaz fan.**





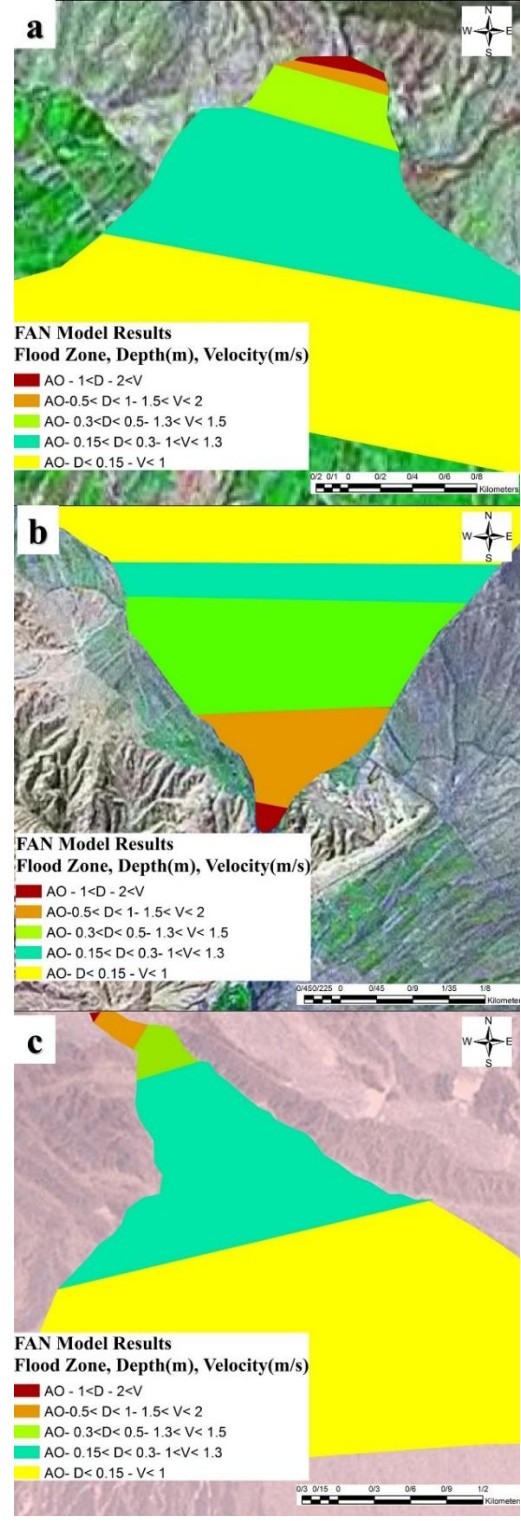

**Figure 8. Flood velocity results of FLO-2D model for: A) Ardak fan, B) Ferizy fan, and C) Sarbaz fan.**


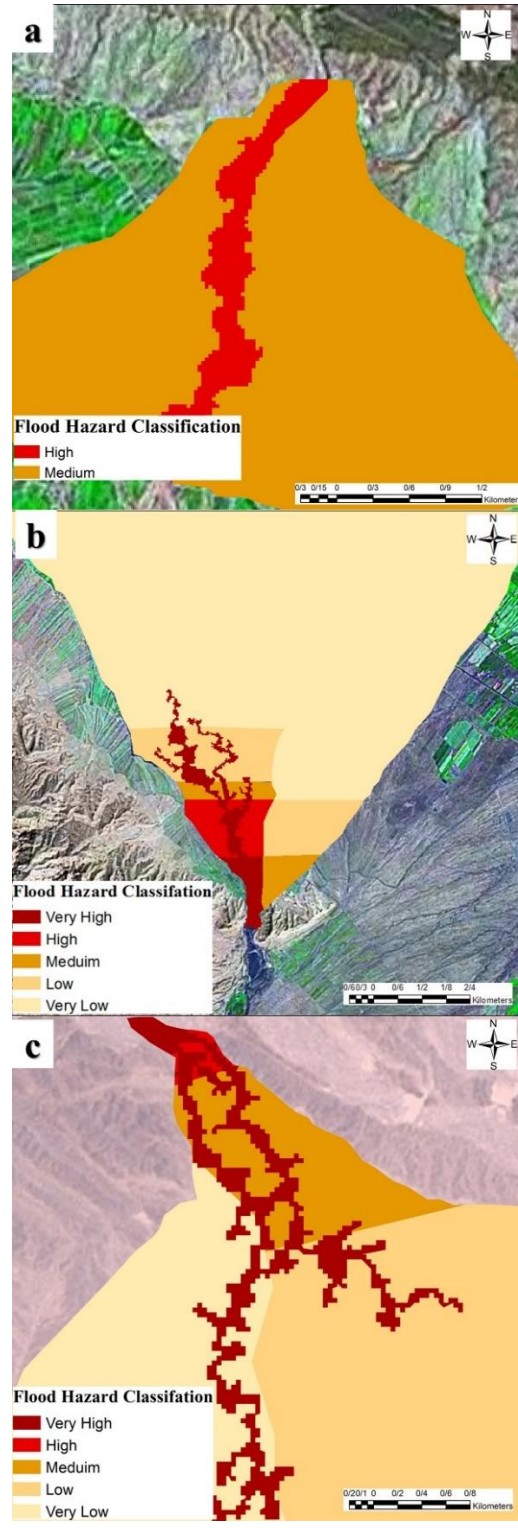

**Figure 9. The final integrated results of FAN, FLO-2D models and geomorphological approach.**