# Peer review of "Enhancing Flood Hazard Estimation Methods on Alluvial Fans Using an Integrated Hydraulic, Geological and Geomorphological Approach"

_Natural Hazards and Earth System Sciences, 2017_

## Referee Comment (RC1) · J. Fuller (Referee) · 25 Jul 2017

This paper addresses an important topic with a unique approach. It is clearly and well written. There are several gaps in the presentation that should be addressed before acceptance for publication. These include the following: 1. It was surprising to see the authors accept use of the FAN methodology as part of their recommended procedure, given their acknowledgement of its many weaknesses. The FAN methodology is outdated and based on bad science. It should no longer be used in modern assessments of flood hazards on alluvial fans. 2. The fan model does NOT predict avulsions (p.

[Figure]

2, line 18). FAN merely assumes that avulsions will always occur, and predicts the probability-weighted depths and velocities of the 1% chance event, i.e., it (incorrectly) predicts the consequences of avulsions. It does not predict whether or where avulsions will occur. According to FAN, avulsions always occur. 3. The article mentions the mapping of geomorphically active (young) and inactive (old) surfaces, but there is no map provided other than the NDVI images. These images are not labelled to distinguish the old and young surfaces. The inadequacies of the FAN method become glaringly apparent when comparing maps of actual floods on fan surfaces, geomorphic surface maps, FLO2D inundation maps (all of which have similar shapes and extents) to FAN model results which stand out in shape, extent, and depth. Enhanced discussion of the surficial mapping element of the study would greatly improve the findings. 4. The article should clarify how the lateral and distal extents of the fan were determined. 5. FLO2D is a useful model for depicting flood hazards on fan surfaces, but the application should include more than a single model with only one hydrograph. The model should be manipulated to account for potential blockages, avulsions, impacts of aggradation, stream piracy, etc.

A few technical issues. 1. p. 1, line 35, the quotation from the NRC report is missing the word "than" between "laid down, [] where deposits..." Verify this on p. 62 of the NRC report. 2. "aggradation" is spelled incorrectly throughout. E.g., p. 2, line 10. 3. FIgure 6 is incorrectly labelled as being FLO2D results
* * *

---

## Referee Comment (RC2) · Anonymous Referee #2 · 8 Aug 2017

[referee-annotated manuscript omitted]

---

## Short Comment (SC1) · 11 Sep 2017

Dear Dr. Fuller,

Special thanks for your illuminating comments on the paper. Sorry for responding rather late, it is due to the fact that some of our colleagues were on summer vacation! You can find the corresponding answers in Supplement as a ZIP file.

With Kind Regards,

The anthers.

Please also note the supplement to this comment:
https://www.nat-hazards-earth-syst-sci-discuss.net/nhess-2017-226/nhess-2017-226-SC1-supplement.zip

---

## Author Comment (AC1) · 5 Nov 2017

Dear Dr. Fuller,

Special thanks for your illuminating comments on our paper. Sorry for responding rather late. We sent the responses on Sept. 11, 2017, however, due to some misunderstanding on our side concerning the submission procedure, the final responses were posted as "short comments". Ms. Anna Feist-Polner, the Editorial Support, guided us to remedy the process. You can find the answers to your comments, and also, the final

version of the manuscript in the supplement as a ZIP file.

Please share with us any further comments you may have. We are sure it will add to the richness of the quality of our scientific approach. Thanks for your time and concern.

With Kind Regards,

S. M. Hasheminia

Please also note the supplement to this comment:
https://www.nat-hazards-earth-syst-sci-discuss.net/nhess-2017-226/nhess-2017-226-AC1-supplement.zip

––––––––––––––––––––––––––––––––

---

## Author Comment (AC2) · 5 Nov 2017

Dear Referee #2

Special thanks for your through review of our paper. Sorry for responding rather late. We sent the responses on Sept. 11, 2017, however, due to some misunderstanding on our side concerning the submission procedure, the final responses were posted as "short comments". Ms. Anna Feist-Polner, the Editorial Support, guided us to remedy the process. All of your comments were considered in the text as precisely as possible.

[Figure]

Please kindly find the corrected manuscript in the attached file and kindly share with us any further comments you may have. We are sure it will add to the richness of the quality of our scientific approach. Thanks for your time and concern.

With Kind Regards,

S. M. Hasheminia

Please also note the supplement to this comment:
https://www.nat-hazards-earth-syst-sci-discuss.net/nhess-2017-226/nhess-2017-226-AC2-supplement.pdf

[Figure]

**Fig. 1.**

[Figure]

[Figure]

[Figure]

**Fig. 2.**

[Figure]

**Fig. 3.**

[Figure]

**Fig. 4.**

[Figure]

1" = 1000'

**Fig. 5.**

[Figure]

**Fig. 6.**

[Figure]

**Fig. 7.**

**Fig. 8.**

**Fig. 9.**

**Fig. 10.**

---

## Author Response (AR1)

**Dear Dr. Fuller**,

Special thanks for your illuminating comments on our paper. Sorry for responding rather late. We sent the responses on Sept. 11, 2017, however, due to some misunderstanding on our side concerning the submission procedure, the final responses were posted as "short comments". Ms. Anna Feist-Polner, the Editorial Support, guided us to remedy the process. You can find the answers to your comments, and also, the final version of the manuscript in the supplement as a ZIP file.

Please share with us any further comments you may have. We are sure it will add to the richness of the quality of our scientific approach. Thanks for your time and concern.

With Kind Regards,

**S. M. Hasheminia**

| Referee: J. Fuller | Email: jon@jefuller.com |
|---|---|

**Question #1.**

It was surprising to see the authors accept use of the FAN methodology as part of their recommended procedure, given their acknowledgement of its many weaknesses. The FAN methodology is outdated and based on bad science. It should no longer be used in modern assessments of flood hazards on alluvial fans.

**Answer to question #1.**

As you mentioned, FAN model is outdated and is based on bad science regarding estimation of the flow depth and velocity. However, In this study, only the probability of flood occurrence from apex was considered from FAN model output. Therefore, to compensate for the model's shortage in respect of velocity and depth calculations, a combination of FAN and FLO-2D was implemented.

**Question #2.**

The fan model does NOT predict avulsions (p. 2, line 18). FAN merely assumes that avulsions will always occur, and predicts the probability-weighted depths and velocities of the 1% chance event, i.e., it (incorrectly) predicts the consequences of avulsions. It does not predict whether or where avulsions will occur. According to FAN, avulsions always occur.

**Answer to question #2.**

The sentence "In addition, it is a simple model which could predict avulsion" has been corrected to: "In addition, it is a simple model which could predict flood risk".

**Question #3.**

The article mentions the mapping of geomorphically active (young) and inactive (old) surfaces, but there is no map provided other than the NDVI images. These images are not labelled to distinguish the old and young surfaces. The inadequacies of the FAN method become glaringly apparent when comparing maps of actual floods on fan surfaces, geomorphic surface maps, FLO2D inundation maps (all of which have similar shapes and extents) to FAN model results, which stand out in shape, extent, and depth. Enhanced discussion of the surficial mapping element of the study would greatly improve the findings.

**Answer to question #3.**

To determine the geomorphological properties, in addition of using field inspections' data and Arial images, vegetation effects of the study area were employed by incorporating the NDVI (Normalized Difference Vegetation Index".

**Question #4.**

The article should clarify how the lateral and distal extents of the fan were determined.

**Answer to question #4.**

To determine the lateral and distal extents of the fan, historical information along with Arial images and elevation maps were considered.

**Question #5.**

FLO2D is a useful model for depicting flood hazards on fan surfaces, but the application should include more than a single model with only one hydrograph.

**Answer to question #5.**

The main objective of this paper was to propose a new approach to enhance flooding hazard in Fans while using minimum data, so only a typical hydrograph was the subject of this study. However, different hydrographs can be considered in future studies for different flooding scenarios.

**Dear Referee #2,**

Special thanks for your through review of our paper. Sorry for responding rather late. We sent the responses on Sept. 11, 2017, however, due to some misunderstanding on our side concerning the submission procedure, the final responses were posted as "short comments". Ms. Anna Feist-Polner, the Editorial Support, guided us to remedy the process. All of your comments were considered in the text as precisely as possible.

Please find the corrected manuscript in the attached file and kindly share with us any further comments you may have. We are sure it will add to the richness of the quality of our scientific approach. Thanks for your time and concern.

With Kind Regards,

**S. M. Hasheminia**

[revised manuscript text omitted]

---

## Referee Report (RR1)

[referee-annotated manuscript omitted]

---

## Author Response (AR2)

**Dear Referee #3;**

Special thank for your detailed and precise review of our manuscript. Please note that all of your valued suggestions and corrections were included in the enclosed final version. In addition, the effectiveness of the new proposed method was discussed further, and the advantages were mentioned. Also, the historical data was added to the manuscript as you requested.

VWith kind regards,

S. M. Hasheminia

[revised manuscript text omitted]

---

## Author Response (AR3)

**Dear Dr. Mario Parise;**

Special thank for your detailed and precise review of our manuscript. Please note that all of your valued suggestions and corrections has been included in the enclosed final version of manuscript both as highlighted and non-highlighted form which is attached as the supplementary material.

With kind regards,

S. M. Hasheminia

[revised manuscript text omitted]